# RBM20 Regulates CaV1.2 Surface Expression by Promoting Exon 9* Inclusion of *CACNA1C* in Neonatal Rat Cardiomyocytes

**DOI:** 10.3390/ijms20225591

**Published:** 2019-11-08

**Authors:** Akihito Morinaga, Jumpei Ito, Tomoaki Niimi, Andrés D. Maturana

**Affiliations:** 1Laboratory of Animal Cell Physiology, Graduate School of Bioagricultural Sciences, Nagoya University, Nagoya 464-8601, Japan; nago.akihito89@gmail.com (A.M.); tniimi@agr.nagoya-u.ac.jp (T.N.); 2The School of Cardiovascular Medicine and Sciences, King’s College London British Heart Foundation Centre of Excellence, London SE5 9NU, UK; jumpei.ito@kcl.ac.uk

**Keywords:** l-type voltage-gated calcium channels, alternative splicing, cardiomyocytes, RBM20

## Abstract

The *CACNA1C* gene encodes for the CaV1.2 protein, which is the pore subunit of cardiac l-type voltage-gated calcium (Ca^2+^) channels (l-channels). Through alternative splicing, *CACNA1C* encodes for various CaV1.2 isoforms with different electrophysiological properties. Splice variants of CaV1.2 are differentially expressed during heart development or pathologies. The molecular mechanisms of *CACNA1C* alternative splicing still remain incompletely understood. RNA sequencing analysis has suggested that *CACNA1C* is a potential target of the splicing factor RNA-binding protein motif 20 (RBM20). Here, we aimed at elucidating the role of RBM20 in the regulation of *CACNA1C* alternative splicing. We found that in neonatal rat cardiomyocytes (NRCMs), RBM20 overexpression promoted the inclusion of *CACNA1C’s* exon 9*, whereas the skipping of exon 9* occurred upon RBM20 siRNA knockdown. The splicing of other known alternative exons was not altered by RBM20. RNA immunoprecipitation suggested that RBM20 binds to introns flanking exon 9*. Functionally, in NRCMs, RBM20 overexpression decreased l-type Ca^2+^ currents, whereas RBM20 siRNA knockdown increased l-type Ca^2+^ currents. Finally, we found that RBM20 overexpression reduced CaV1.2 membrane surface expression in NRCMs. Taken together, our results suggest that RBM20 specifically regulates the inclusion of exon 9* in *CACNA1C* mRNA, resulting in reduced cell-surface membrane expression of l-channels in cardiomyocytes.

## 1. Introduction

l-type voltage-gated calcium (Ca^2+^) channels drive the excitation–contraction coupling in cardiac myocytes [1,2]. CaV1.2 (α1) protein forms the Ca^2+^ selective filter pore of l-type voltage-gated Ca^2+^ channels. Auxiliary subunits (β, α2σ, and γ) regulate the channel’s gating and plasma membrane targeting [2]. The *CACNA1C* gene encodes for the CaV1.2 subunit in cardiac myocytes. The *CACNA1C* human gene contains 50 exons, among which six exons of *CACNA1C* are subject to alternative splicing (Figure 1). The alternative splicing of *CACNA1C* regulates CaV1.2 tissue-specific expression, alters pharmacological sensitivity, and is associated with pathological cardiac syndromes [3]. Different splicing factors regulate alternatively spliced exons of *CACNA1C*. Exon 9* skipping and exon 33 inclusion are regulated by the RNA-binding fox family proteins (Rbfox1 and RBfox2) [4,5]. The polypyrimidine tract-binding protein (PTBP1) regulates the splicing switch between exons 8 and 8a in the brain and possibly promotes exon 8a expression in the heart [6]. 

The splicing factor RNA-binding motif 20 (RBM20) is a regulator of alternative splicing and plays a role in heart pathophysiology [7,8]. mRNA sequencing of an RBM20 knockout (KO) mouse model revealed that RBM20 could regulate the splicing of exons 8, 9*, 22, and 31 of *CACNA1C* [9]. RBM20 is highly expressed in the heart and mutations in RBM20 cause familial dilated cardiomyopathy (DCM) by promoting the expression of a long isoform of the sarcomere protein titin [10]. 

Here, we examined the role of RBM20 in the alternative splicing of *CACNA1C* in neonatal rat cardiomyocytes (NRCMs). We found that alteration of RBM20 expression, by either overexpression or siRNA knockdown, specifically targets exon 9*. RBM20 overexpression resulted in a reduction of l-type voltage-gated Ca^2+^ currents, whereas RBM20 siRNA knockdown increased l-type voltage-gated Ca^2+^ currents. RNA immunoprecipitation (IP) showed that RBM20 binds to exon 9*, flanking introns that contains three conserved consensus sequences for RBM20. Finally, we found that RBM20 modulates the membrane surface expression of CaV1.2, suggesting that exon 9* plays a role in the targeting of CaV1.2 in the plasma membrane in NRCMs. Taken together, our results suggest that RBM20 regulates the inclusion of exon 9*, thus altering the membrane surface expression of CaV1.2.

## 2. Results

### 2.1. RBM20 Promotes CACNA1C Exon 9* Inclusion in Cardiomyocytes 

An RNA sequencing analysis showed that *CACNA1C* is a target for RBM20 for alternative splicing [9]. Exons 8, 9*, 22, and 31 were identified as potential targets for RMB20 splicing regulation. To confirm the regulation of these exons by RBM20, we manipulated RBM20 expression and tested the effects of the overexpression of RBM20 in cultured NRCMs, which was performed by adenoviral transduction at a multiplicity of infection (MOI) of 10 (Figure 1B). MOI up to 100 has been previously used and showed no morphological or toxic effects on cardiomyocytes [11]. RBM20 was efficiently knocked down by a specific siRNA (Figure 1B). Amplified cDNAs resulting from RT-PCR were loaded on 8% polyacrylamide gel (PAGE) and stained with ethidium bromide after electrophoresis. Primers were designed to amplify regions from exons 1 to 4; 1a to 4; 8 to 9; 8a to 9; 7 to 10; 20 to 22; between 19, 20, and 22; and from 30 to 35 (Figure 2A and Appendix A). The RT-PCR for exon regions 7–10 showed two amplicon bands of 441 base pairs (bp) and 516 bp. The amplicon of the 441 bp band corresponded to the exon regions 7–10. The 516 bp band corresponded to exons 7–10, including exon 9*. In the RBM20 overexpressing condition (Figure 2A, left), the band was found to be more intense than the control and Green Fluorescent protein (GFP) overexpression. In contrast, upon RBM20 siRNA knockdown (Figure 2A, right), the band at 516 bp was less intense than in the control and luciferase-1-targeting siRNA (siLuc) conditions. As a control, expression of the titin long isoform was also increased by RBM20 siRNA knockdown (Appendix A).

To confirm RT-PCR results, we measured by quantitative RT-PCR the amount of exon 9* (Figure 2B). Exon 9* was significantly increased when RBM20 was overexpressed compared with the control and GFP overexpression. In contrast, siRNA knockdown of RBM20 significantly reduced the amount of exon 9*. In both RBM20 overexpression and siRNA knockdown conditions, we did not observe any changes for the other tested alternative exons (Appendix A). These results thus suggest that RBM20 specifically regulates exon 9* inclusion of *CACNA1C*.

### 2.2. RBM20 Binds the Intronic Region Surrounding Exon 9* 

The consensus sequence on mRNA for RBM20 binding is a UCUU motif [12]. We found two conserved RBM20 consensus motifs, among humans, mice, and rats, in the intron upstream of exon 9* and one in the intron downstream (Figure 3A). Therefore, we tested whether RBM20 could bind the flanking introns of exon 9* by RNA IP (Figure 3B). In NRCMs, the immune precipitate of FLAG-RBM20, an amplicon, was found to correspond to the 3′ intronic region of exon 9* to the 5′ intronic region. Western blotting was performed to verify the expression of FLAG-RBM20 in immunoprecipitated samples. These results thus suggest that RBM20 binds to the introns flanking exon 9*.

### 2.3. Manipulation of RBM20 Expression Alters l-tType Ca^2+^ Currents in Cardiomyocytes

To test whether RBM20 affects l-type voltage-gated Ca^2+^ channel activity, we next measured l-type Ca^2+^ currents in isolated NRCMs by a patch clamp in the whole-cell configuration.

l-type Ca^2+^ currents were elicited by 10 mV depolarization steps from −60 to +70 mV from a holding potential at −40 mV. RBM20-overexpressing cells showed a significantly smaller l-type Ca^2+^ current density than the control or GFP-overexpressing cells between 0 and +30 mV (Figure 4A). 

We next measured l-type Ca^2+^ currents upon RBM20 siRNA knockdown. In RBM20 knockdown cells, l-type Ca^2+^ currents were significantly increased compared with control and siLuc (Figure 4B). The Ca^2+^ current density increased significantly at +20 and +30 mV.

These results suggest that the RBM20 regulation of exon 9* splicing plays a role in either the expression or the activity of l-type voltage-gated calcium channels in NRCMs. 

### 2.4. RBM20 Represses the Membrane Surface Expression of CaV1.2

RBM20 modulation of l-type calcium currents suggests a possible regulatory role in the expression of CaV1.2. To test this hypothesis, we estimated the protein amount of CaV1.2. We first performed Western blotting using whole-cell protein lysates of cardiomyocytes in control, GFP, or RBM20 overexpression conditions (Figure 5A). The expression of CaV1.2 was slightly reduced compared with GFP overexpression and control but without any significant effect. We next verified whether RBM20 affects the membrane surface expression of CaV1.2 by membrane protein biotinylation. The protein amount of CaV1.2 in the membrane was found to be significantly reduced compared with the control and GFP conditions. These results thus suggest that RBM20 regulates CaV1.2 membrane surface expression possibly through exon 9* inclusion.

## 3. Discussion

We report here that, in NRCMs, the splicing factor RBM20 promotes the inclusion of exon 9* of the *CACNA1C* gene encoding for CaV1.2, the pore subunit of l-type voltage-gated Ca^2+^ channels. Furthermore, our results show that manipulation of RBM20 expression affects l-type Ca^2+^ currents in rat neonatal ventricular myocytes, as well as the membrane expression of CaV1.2 at the plasma membrane surface, suggesting that RBM20 plays a role in the plasma membrane targeting of CaV1.2 possibly through the regulation of exon 9* inclusion. 

An RNA sequencing analysis and calculation of percent spliced-in score (PSI) revealed a small score of *CACNA1C* exon 8, 9*, 22, and 31 splicing events [9]. Using NRCMs, we confirmed that exon 9* expression was altered upon RBM20 overexpression or downregulation by siRNA knockdown. However, other known alternatively spliced exons were not affected by RBM20 overexpression or knockdown. This suggests that RBM20 specifically target the exon 9* of *CACNA1C* in NRCMs. Although RNA sequencing is a powerful method for RNA analysis, obtaining a false positive in the exon–exon junction is a known problem that is currently being addressed [13,14]. Therefore, further confirmation by molecular methods such RT-PCR remain useful. 

Here, our results show that manipulation of RBM20 expression suggests that RBM20 plays a role in exon 9* inclusion (Figure 2). A previous study showed that the splicing factor Fox1 (Feminizing on the X 1, also called A2BP1) regulates exon 9* exclusion [4]. Thus, it is possible that Fox1 and RBM20 oppositely regulate, respectively, skipping and inclusion of exon 9* in *CACNA1C*. 

CaV1.2 isoforms, including exon 9*, are highly expressed in smooth muscle, while their expression is low in the heart ventricle. However, in myocardial infraction model rats, surviving cardiomyocytes in scar regions expressed high levels of CaV1.2, including exon 9* [15]. 

When exon 9*–CaV1.2 was expressed in HEK293 cells together with the β2A subunit, the only difference with wild-type CaV1.2 was a hyperpolarization shift of about 10 mV of the current–voltage relationship and voltage-dependent activation [16]. We did not observe this hyperpolarization shift in isolated NRCMs in the unit range of 10 mV, since there is a possibility that a channel activation shift may occur in smaller voltage units in cardiomyocytes. 

We found that upon RBM20 overexpression, l-type Ca^2+^ currents were reduced, whereas upon RBM20 siRNA knockdown, l-type Ca^2+^ currents were increased. In the vicinity of exon 9*, there is a binding site for the β subunit [17]. The inclusion of exon 9* could, therefore, alter the interaction with the β subunit or promote the binding of a different β subunit type. It is known that the β subunit is essential for CaV1.2 targeting of the plasma membrane [18]. It is therefore possible that exon 9* inclusion affects CaV1.2 membrane targeting by affecting the β subunit binding, a hypothesis that should be tested. We also cannot rule out that other protein alternative splice variants regulated by RBM20 might affect the membrane targeting of CaV1.2. 

A limitation of our present study is that we only used two extreme conditions of RBM20 expression: Overexpression and silencing. We chose these conditions in order to clearly identify the exon in *CACNA1C* regulated by RBM20. Further manipulation of milder RBM20 overexpression and siRNA knockdown should be performed. Similar to siRNA knockdown, a significant reduction of RBM20 has been previously measured in cardiomyocytes treated with neurohumoral factors known to stimulate cardiac hypertrophy [11]. It would be thus interesting to now investigate the physiological significance of *CACNA1C* splicing by RBM20. 

In conclusion, our study revealed that in NRCMS, RBM20 regulates the surface expression of the CaV1.2 subunit of the l-type Ca^2+^ channel possibly by specifically regulating the inclusion exon 9* of *CACNA1C*. 

## 4. Materials and Methods

### 4.1. Experimental Animals

The care and handling of rats for the present study were in conformity with the Animal Research Advisory Committee Guidelines (on-line: https://oacu.oir.nih.gov/animal-research-advisory-committee-guidelines) and approved by the Animal Care and Use Committees of Nagoya University (elucidating the mechanisms of gene expression in cardiac diseases, 2018031374, 30 March 2018).

### 4.2. Cell Culture

The hearts from sacrificed three-day-old neonatal Wistar/ST rats were removed after dissection. To isolate cardiomyocytes, the cardiac muscle tissue was digested in 10 mL of phosphate-buffered saline (PBS) containing 0.12% type 2 collagenase at 37 °C and pelleted by centrifugation at 950 rpm for 5 min. The pellet, containing cells, was then suspended in Dulbecco’s modified Eagle’s medium with 10% FBS, 1% penicillin–streptomycin, and 100 nM BrDU five times. Cells were then plated on 10 cm dishes for 1 h to separate cardiomyocytes from fibroblasts. Finally, cardiomyocytes were cultured in Dulbecco’s modified Eagle’s medium with 10% FBS, 1% penicillin–streptomycin, and 100 nM BrDU in 5% CO_2_ at 37 °C for 24 h. The culture medium was changed to Dulbecco’s modified Eagle’s medium containing 1% penicillin-streptomycin, 100 nM BrDU, 1 µg/mL of insulin, and 5 µg/mL of transferrin.

After 24 h, cells were infected with RBM20-adenovirus or GFP-adenovirus as a negative control. We transduced NRCMs at a MOI of 10, MOI up to 100 has shown no toxic side effects on cells [11]. Transfection was done with siRNA-targeting RBM20 (siRBM20) or luciferase-1-targeting siRNA [11] as a negative control using Lipofectamine RNAiMAX transfection reagent (Invitrogen, Waltham, Massachusetts, USA) at 20 nM, as previously described [11]. 

### 4.3. RNA Isolation, Reverse Transcription, PCR, and Electrophoresis

Total RNA from cardiomyocytes was extracted using TRIreagent (SIGMA-ALDRICH, ST. Louis, MO, USA) according to the manufacturer’s indications. Total RNA was reverse-transcribed using the ReverTra Ace qPCR RT Kit (Toyobo, Tokyo, Japan). PCR was performed with the synthesized cDNA (50 ng of RNA content), 1 µL of primers (10 µM), and EmeraldAmp^®^ PCR Master Mix (TaKaRa, Shiga, Japan) in a final volume of 25 µL. The sequence of the primers is indicated in Table 1. We used 8% PAGE electrophoresis to separate and visualize the amplicons.

### 4.4. Real-Time RT-PCR

RT-PCR was performed with the synthesized cDNA (20 or 10 ng of RNA content), 8 µL of primers (1 µM), and THUNDERBIRD SYBR qPCR Mix (Toyobo, Tokyo, Japan) in a final volume of 20 µL. Applied Biosystems 7300 Real-Time PCR System (life technologies, Waltham, MA, USA,) was used for the PCR reaction. The mRNA levels were normalized to GAPDH with the 2^−∆∆Ct^ method. The sequence of the primers is indicated in Table 1.

### 4.5. Cell-Surface Biotinylation, Protein Isolation, and Western Blot Analysis

Proteins were isolated from NRCMs with a lysis buffer (1% NP-40, 150 mM NaCl, and 50 mM Tris-HCl) and protease inhibitor cocktail tablets (Roche, Mannheim, Germany).

Cell-surface proteins biotinylated and isolated from NRCMs were extracted using the Piece Cell-Surface Protein Isolation Kit (Pierce, Rockford, IL, USA) according to the manufacturer’s indications. For elution, we used an SDS-PAGE sample buffer (62.5 mM Tris-HCl (pH 6.8), 1% SDS, 10%)

Cell lysates were resolved by 7.5% SDS-PAGE electrophoresis for the detection of α1C, GAPDH, and ATP1B1. Proteins were transferred to a polyvinylidene difluoride membrane (Millipore, USA). The respective proteins were detected with anti-α1C (1/750) polyclonal antibody (Alomone Labs, Jerusalem, Israel), anti-GAPDH (1/2000) polyclonal antibody (MBL, Nagoya, Japan), and anti-ATP1B1 (1/750) polyclonal antibody (Proteintech, Rosemont, IL, USA). The protein concentrations were normalized to GAPDH. Cell-surface protein concentrations were normalized to ATP1B1. Chemiluminescence was detected using the ELC Plus Western Blotting System (GE Healthcare Chicago, IL, USA) and the LAS-1000 plus luminescent image analyzer (Fujifilm, Tokyo, Japan).

### 4.6. Patch Clamp

Cardiomyocytes were plated on a poly-l-lysine-coated glass bottom dish. Patch clamp micropipettes made from borosilicate glass capillaries (0.86 mm diameter) (Harvard apparatus, Holliston, MA, USA) were fabricated using a puller (PC-10, Narishige, Toyko, Japan) and fire polished with a microforge (MF-900, Narishige, Tokyo, Japan Japan). 

The bath solution contained 125 mM N-methyl-D-glucamine, 5 mM 4-AP, 20 mM TAECl, 2 mM CaCl_2_, 10 mM D-glucose, and 10 mM HEPES (pH 7.4 with TEAOH). The pipette solution contained 130 mM CsCl, 10 mM EGTA, 25 mM HEPES, 3 mM Mg-ATP, and 0.4 mM Li-GTP (pH 7.2 with CsOH). Cells were placed under an inverted microscope (Olympus IX71) and whole-cell Ca^2+^ currents were measured in the voltage clamp mode using an Axopatch 200B amplifier (Molecular Devices, St. Jose, CA, USA). The cells were held at a membrane potential of −40 mV and depolarized by 10 mV voltage steps of up to +60 mV to measure l-type Ca^2+^ currents. The leak was subtracted automatically by the Clamp 10 software (Molecular Devices, St. Jose, CA, USA).

### 4.7. Data Analysis

Statistical analysis was performed by unpaired Student’s *t*-tests. Data are presented as mean ± SEM (* means *p* < 0.05, ** means *p* < 0.01, and *** means *p* < 0.001 compared to control samples).

## Figures and Tables

**Figure 1 ijms-20-05591-f001:**
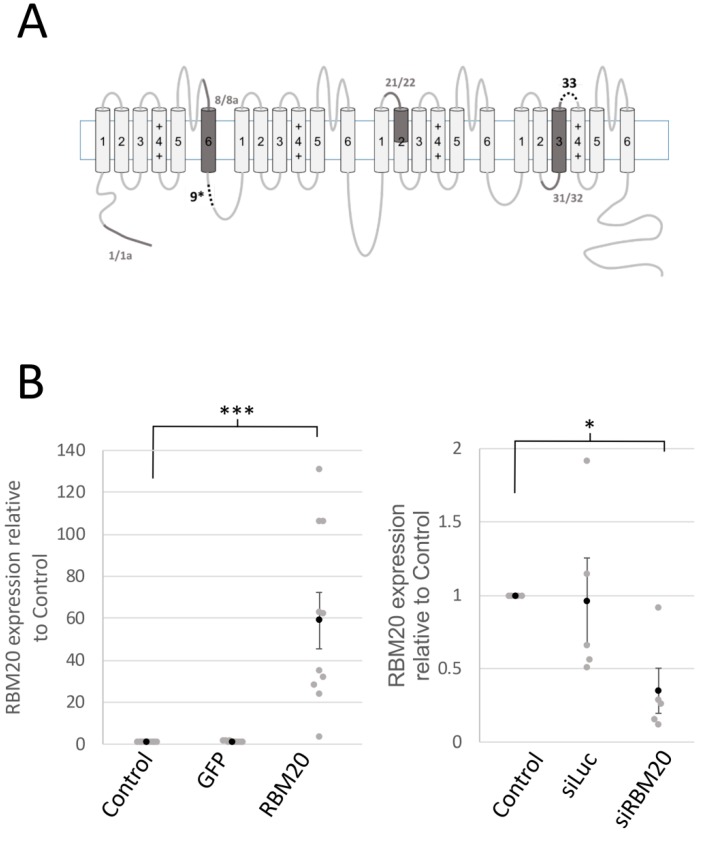
The splicing factor RNA-binding motif 20 (RBM20) regulates the inclusion of alternatively spliced exon 9*. (**A**) The scheme represents the α1C subunit of l-type voltage calcium channels. CaV1.2 (α1C) is composed of four homologous regions consisting of six transmembrane domains (S1–S6) that form the pore of the channel. (+) indicates voltage sensors on S4. There are two types of alternatively spliced exons on CaV1.2: [1] Mutually exclusive exons—1/1a, 8/8a, 21/22, and 31/32 (the corresponding region is shown in dark)—and [2] optional exons—9* and 33 (shown as dotted lines). (**B**) The graphs show RBM20 expression measured by quatitative RT-PCR. The left graph is the expression of RBM20 measured in the control, Green Fluorescent protein (GFP)-overexpressing, and RBM20-overexpressing cardiomyocytes. Data are the mean and SEM (black circle) of *n* = 10 total RNA extracts from 10 cardiomyocyte isolations (gray circle). The right graph shows the expression in the control, luciferase-1-targeting siRNA (siLuc) tranfected cardiomyocytes, and siRNA-targeting RBM20-overexpressing cardiomyocytes. Data are the mean and SEM (black circle) of *n* = 5 total RNA extracts from 5 cardiomyocyte isolations (gray circle). * means *p* < 0.05 and *** *p* < 0.001.

**Figure 2 ijms-20-05591-f002:**
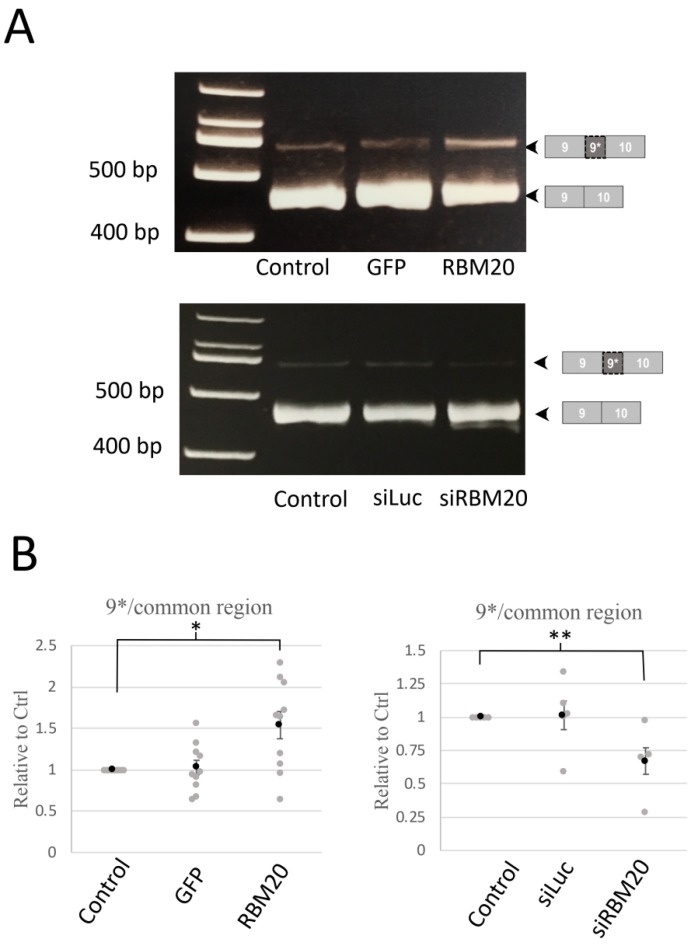
Manipulation of RBM20 expression modulates exon 9* splicing. Splicing of exon 9* was monitored by RT-PCR and RT-qPCR amplification of cardiomyocyte total mRNA. (**A**) Pictures of RT-PCR reaction resolved on an 8% PAGE gel. Amplicons are a region between exons 9 and 10, including or excluding exon 9* in the control, GFP, and RBM20-overexpressing cardiomyocytes (**upper**) or in the control, luciferase-1-targeting siRNA (siLuc), and siRNA-targeting RBM20 (siRBM20)-treated cardiomyocytes (**lower**). Pictures are representative of RT-PCR experiments from total RNA preparation extracted from neonatal rat cardiomyocytes (*n* = 5). (**B**) Graphs show the ratio of exon 9*, including the region over common regions (exons 19–20), of quantitative RT-PCR amplification in control, GFP, and RBM20-overexpressing cardiomyocytes (**left**, *n* = 10) or luciferase-1-targeting siRNA (siLuc), and siRNA-targeting RBM20 (siRBM20)-treated cardiomyocytes (right *n* = 5). Data are mean and SEM (gray circle) and individual data (gray circle); ** *p* < 0.01, * *p* < 0.05.

**Figure 3 ijms-20-05591-f003:**
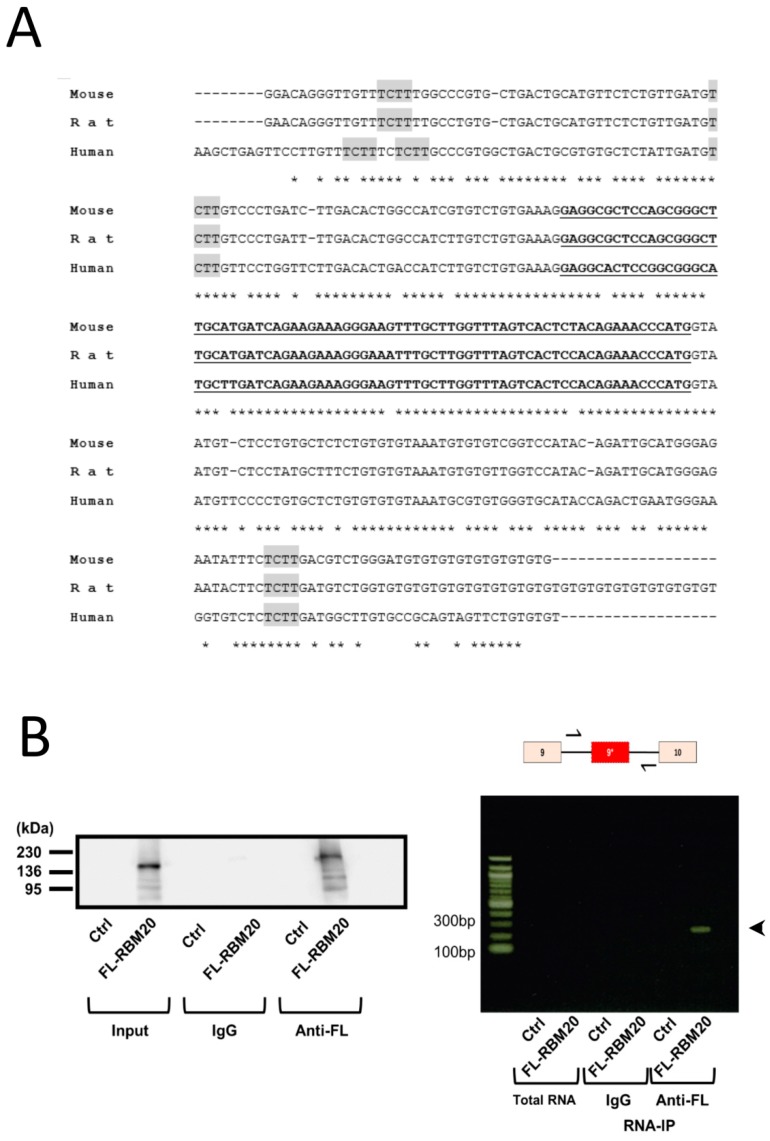
RBM20 bound to introns surrounding exon 9*. (**A**) Sequence allignment of exon 9* (bold and underlined) and partial flanking introns of exon 9* from mice, rats, and humans. Three conserved putative RBM20 binding sites (TCTT) in all three species are highlighted in gray. Stars under the sequences indicate inadequate sequence homology between the tree species. (**B**) RNA IP expression of FLAG-tagged RBM20 overexpressed in neonatal rat cardiomyocytes. Left is a picture of a Western blot showing FLAG-tagged RBM20 expression of immunoprecipitated extracts with IgG or anti-FLAG M2 antibodies. Right is a picture of agarose gel from an RT-PCR amplicon resulting from RNA IP. Upper scheme shows the exon–intron region from exon 9 to 10, with arrows indicating the forward and reverse primers used in this experiment. Pictures are representative of three independent RNA IPs from three different cardiomyocyte preparations.

**Figure 4 ijms-20-05591-f004:**
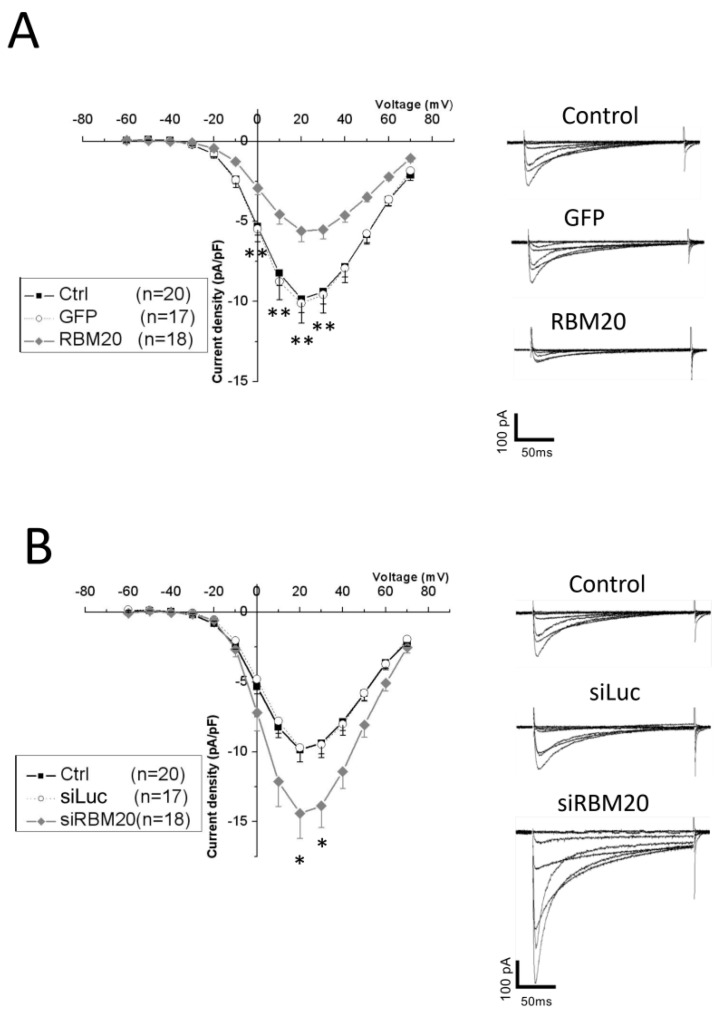
Manipulation of RBM20 expression altered l-type calcium currents in cardiomyocytes. (**A**) l-type Ca^2+^ current-density—voltage curves recorded by patch clamp in the whole-cell configuration, from a −40 mV holding potential with a 10 mV voltage step ranging from −60 to +70 mV. Currents were measured in control (black square), GFP (open circle), and RBM20-overexpressing (gray diamond) cardiomyocytes. The number of measured cells is indicated in the figure inset. Data are mean and SEM; ** is *p* < 0.01. Recorded traces on the right show an example of recorded calcium current traces in control (ctrl), GFP, and RBM20-overexpressing cardiomyocytes with current and time scale bars. (**B**) l-type Ca^2+^ current-density–voltage curves recorded by patch clamp in the whole-cell configuration, from a −40 mV holding potential with a 10 mV voltage step ranging from −60 to +70 mV. Currents were measured in control (black square), luciferase-1-targeting siRNA (open circle), and RBM20 siRNA (gray diamond) cardiomyocytes. The number of measured cells is indicated in the figure inset. Data are mean and SEM; * is *p* < 0.05. Recorded traces on the right show an example of the recorded calcium current traces in control (ctrl), luciferase-1-targeting siRNA (siLuc), and RBM20 siRNA (siRBM20) with current and time scale bars.

**Figure 5 ijms-20-05591-f005:**
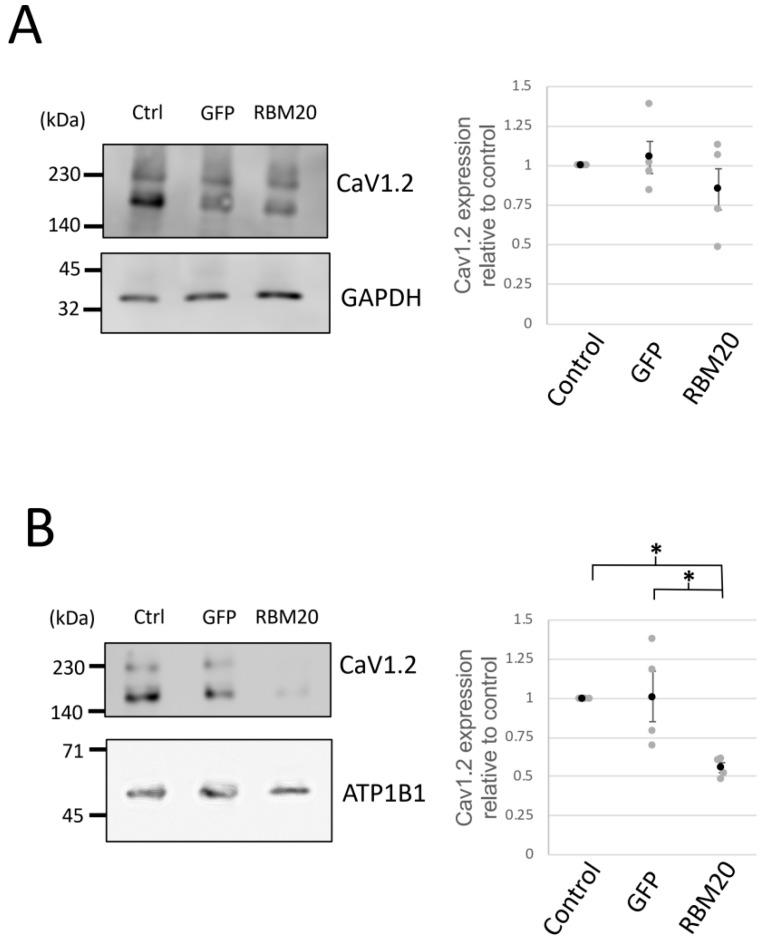
RBM20 modulates CaV1.2 membrane surface expression. (**A**) CaV1.2 protein expression levels were assesed by Western blot experiments. Pictures (**left**) of a whole-cell protein extract Western blot experiment showing CaV1.2 (**upper**) and GAPDH (**lower**) expression in control (ctrl), GFP, and RBM20-overexpressing cardiomyocytes. Graph (**right**) shows the ratio of the band intensity quantification from CaV1.2 over GADPH. Data are mean and SEM (black circle) from *n* = 4 different cardiomyocyte protein extracts (gray circle). (**B**) Cav1.2 membrane surface expression was assayed by membrane protein biotinylation followed by Western blot. Pictures show CaV1.2 (**upper**) and ATP1B1 (**lower**) expression. The graph shows the ratio of band intensity quantification from CaV1.2 over ATP1B1. Data are mean and SEM (black circle) from *n* = 4 different cardiomyocyte protein extracts (gray circle); * is *p* < 0.05.

**Table 1 ijms-20-05591-t001:** Primers used in real-time quantitative PCR and RT-PCR. The table shows the sequences of the primer pairs used for the real-time quantitative PCR and RT-PCR experiments in the present study. Fw: Forward primer; Rev: Reverse primer.

*Gene*	Oligonucleotide Sequences
*GAPDH*	Fw: 5′-CAACTCCCTCAAGATTGTCAGCAA-3′
	Rev: 5′-GGCATGGACTGTGGTCATGA-3′
*RBM20*	Fw: 5′-GGCTTACACAGAAGCTGCTCAAG-3′
	Rev: 5′-GTGGATGTCCTGGATGATAGCAG-3′
*Titin*	Fw: 5′-CAGGAGCAGGTTTCTTTGGA-3′
	Rev: 5′-GAGCCGTATGAGGAACCGTA-3′
*CACNA1C Exon 1a–4*	Fw: 5′-AGCGATAAGGCCGTATGAGA-3′
	Rev: 5′-GTTCCAGGTTGGAGTTGGTG-3′
*CACNA1C Exon 1–4*	Fw: 5′-CAATGGTCAATGAAAACACGA-3′
	Rev: 5′-GTTCCAGGTTGGAGTTGGTG-3′
*CACNA1C Exon 7–10*	Fw: 5′-CACCAACTTCGACAACTTCG-3′
	Rev: 5′-CCACAGTTTTCACCCTCGAT-3′
*CACNA1C Exon 8–9*	Fw: 5′-GGGTCAATGATGCCGTAGGAA-3′
	Rev: 5′-TCTCCTCGAGCTTTGGCTTTC-3′
*CACNA1C Exon 8a–9*	Fw: 5′-ATGCAAGACGCTATGGGCTAT-3′
	Rev: 5′-TCTCCTCGAGCTTTGGCTTTC-3′
*CACNA1C Exon 19–21*	Fw: 5′-GCCGGAAGCCAGTGCATTTT-3′
	Rev: 5′-AACAATGTCAAAATAAAACAGA-3′
*CACNA1C Exon 19–22*	Fw: 5′-GCCGGAAGCCAGTGCATTTT-3′
	Rev: 5′-ATAGTCTGCATTGCCTAGGAT-3′
*CACNA1C Exon 30–35*	Fw: 5′-GCCTCTTCACGGTGGAG-3′
	Rev: 5′-TCCCAATCACTGCATAGATAA-3′

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
