# Peer review of "RBM20 Regulates CaV1.2 Surface Expression by Promoting Exon 9* Inclusion of CACNA1C in Neonatal Rat Cardiomyocytes"

_ijms, 2019, doi:10.3390/ijms20225591_

Round 1

Reviewer 1 Report

Alternative splicing is an important process during gene expression which allows diversifying structure and function of proteins encoded in a single gene. Regulation of splicing is crucial for fine-tuning of cellular functions during development and pathologies. In this study, Akihito et al. explore the role of the RNA Binding Protein Motif 20 in alternative splicing of the cacna1C gene in neonatal rat cardiomyocytes. The authors used well established biochemical and physiological techniques to detect particular splicing events and its consequences on the protein's function. Overall the experiments are well designed and performed, and I agree with most of the conclusions made in the manuscripts. However, I have several major concerns and comments that have to be very carefully addressed before this manuscript can be published.

I think this study is quite limited in terms of tested conditions and possible assays performed. For example, the authors tried only two extreme conditions of the RBM20 expression, either overexpression or silencing, and it is not quite clear how much these conditions are physiologically relevant. The authors have to justify and explain why these conditions were selected and how they are biologically relevant. I would strongly recommend, no matter what justification the authors have, that additional levels of RBM20 expression are tested. It can be easily achieved using different titers of adenovirus and different amounts of siRNA delivered to the cells via transfection. I am surprised that the authors did not perform other simple experiments to assess cells under different conditions of splicing. For example, if cell morphology or cell health were affected by overexpression or suppression of RBM20 proteins. These are simple experiments and should be done during revision. Another big concern is data presentation in the text. It is already well accepted in the field that individual data points have to be shown if less than 10. Please replot all plots to show individual data points if n<10. Figure 2A, the images of gels have different backgrounds, please show raw images for both gels. Figure 4A and 4B have identical figure legends but different plots, which is correct? The manuscript has lots of typos and errors, please do extensive editing. A few examples, line 46 “is represents”; lots of missing commas, lines 182-183, many confusing articles. The authors used NIH Guide published in 1996! NIH updates rules for animal care almost every year, please use the latest NIH Guide for animal care.

Author Response

We thank the reviewer for her/his constructive comments.

Our reply is in the attached file.

Reviewer 2 Report

This study conducted by Akihito and colleagues reported the RBM20 regulate the inclusion of exon 9* in cacna1C mRNA and reduce the L type voltagegated calcium channedls express in the cell membrane but not in the whole cell. The author use PCR , Western blot and whole cell recording to verify the hypothesis. The conclusion is clear and convincing, While the molecular mechanism findings are interesting, several issue should be addressed.

1 In the RBM 20 overexpress experiment. What is the “Control group” mean, and also in the RBM 20 siRNA expriment, What is the control siRNA means?

2 The western blot result about the levels of CaV1.2 in the membrane and whole protein Fig 5 A and B, the numbers are n= 3 and 4. The number of each group should be added  to at least 6.

Author Response

Reviewer 2

We thank the reviewer for her/his constructive comments and suggestions.

This study conducted by Akihito and colleagues reported the RBM20 regulate the inclusion of exon 9* in cacna1C mRNA and reduce the L type voltagegated calcium channedls express in the cell membrane but not in the whole cell. The author use PCR , Western blot and whole cell recording to verify the hypothesis. The conclusion is clear and convincing, While the molecular mechanism findings are interesting, several issue should be addressed.

1 In the RBM 20 overexpress experiment. What is the “Control group” mean, and also in the RBM 20 siRNA expriment, What is the control siRNA means?

The control group means WT cardiomyocytes cells without any treatment and the control siRNA is a siRNA targeting Luciferase1 gene. We corrected from siCtrl to siLuc in methods section and in figures and in legends.

2 The western blot result about the levels of CaV1.2 in the membrane and whole protein Fig 5 A and B, the numbers are n= 3 and 4. The number of each group should be added  to at least 6.

In figure 5, we perform western-blots for total protein and membrane biotinylation on 4 biological replicates. Although we agree with the reviewer that increasing the biological replicates is better, however 3-4 replicates are generally acceptable (Figure 10 in Gao S. et al. J. Mol. And Cellular Cardiology 2019, 137:59-70; Figure 4 in Caceres PS et al. PNAS 2019,116:11796-11805). Western Blot from the total protein was 4 and not as written 3. We corrected the figure legend.

Unfortunately, within 10 days that we have for revising our manuscript, we cannot perform 2 more biological replicates. As suggested by reviewer 1, we show now each individual replicates of the quantification graph.

We thank again the reviewer for her/his helpful comments that we believe improved our manuscript.

Round 2

Reviewer 1 Report

The authors did extensive text editing and figure reformating to address my comments and concerns. Although the authors have not performed any extra experiments for the revision they provided a reasonable explanation for their current experimental design. The only one thing I noticed that has to be corrected is still Figure 4B legend, it states RMB20 overexpression, not siRMB20. Please, correct. Otherwise, I do not have any additional comments and looking forward to reading a follow-up study elucidating the dose-dependent contribution of RMB20 to alternative splicing. 

Author Response

To reviewer 1.

We corrected the mistake of figure 4b legend (Page 4 ligne 153).

We will keep studying exon9* splicing regulation of CACNA1C by RBM20 and we would like to thank the reviewer for his positive comments and usful suggestions on our manuscript and the revision.